# Pentafluorophenyl Platinum(II) Complexes of PTA and Its N-Allyl and N-Benzyl Derivatives: Synthesis, Characterization and Biological Activity

**DOI:** 10.3390/ma12233907

**Published:** 2019-11-26

**Authors:** Paolo Sgarbossa, Urszula Śliwińska-Hill, M. Fátima C. Guedes da Silva, Barbara Bażanów, Aleksandra Pawlak, Natalia Jackulak, Dominik Poradowski, Armando J. L. Pombeiro, Piotr Smoleński

**Affiliations:** 1Dipartimento di Ingegneria Industriale and CIRCC, Consorzio Interuniversitario per le Reattività Chimiche e la Catalisi, Università di Padova, via Marzolo 9, 35131 Padova, Italy; 2Department of Analytical Chemistry, Faculty of Pharmacy, Wrocław Medical University, Borowska 211 A, 50-566 Wrocław, Poland; urszula.sliwinska-hill@umed.wroc.pl; 3Centro de Química Estrutural, Instituto Superior Técnico, Universidade de Lisboa, Av. Rovisco Pais, 1049-001 Lisboa, Portugal; fatima.guedes@tecnico.ulisboa.pt (M.F.C.G.d.S.); pombeiro@tecnico.ulisboa.pt (A.J.L.P.); 4Department of Veterinary Microbiology, Wrocław University of Environmental and Life Sciences, Norwida 31, 50-375 Wrocław, Poland; barbara.bazanow@upwr.edu.pl (B.B.); natalia.jackulak@upwr.edu.pl (N.J.); 5Department of Biochemistry, Pharmacology and Toxicology, Wrocław University of Environmental and Life Sciences, Norwida 31, 50-375 Wrocław, Poland; aleksandra.pawlak@upwr.edu.pl; 6Department of Biostructure and Animal Physiology, Wrocław University of Environmental and Life Sciences, Kożuchowska 1/3, 51-631 Wrocław, Poland; dominik.poradowski@upwr.edu.pl; 7Faculty of Chemistry, University of Wroclaw, F. Joliot-Curie 14, 50-383 Wrocław, Poland

**Keywords:** platinum(II) pentafluorophenyl complexes, N-alkylated PTA, biological activity, HSA interactions

## Abstract

From the well-known 1,3,5-triaza-phosphaadamantane (PTA, **1a**), the novel N-allyl and N-benzyl tetrafuoroborate salts 1-allyl-1-azonia-3,5-diaza-7-phosphaadamantane (APTA(BF_4_), **1b**) and 1-benzyl-1-azonia-3,5-diaza-7-phosphaadamantane (BzPTA(BF_4_), **1c**) were obtained. These phosphines were then allowed to react with (Pt(μ-Cl)(C_6_F_5_)(tht))_2_ (tht = tetrahydrothiophene) affording the water soluble Pt(II) complexes *trans-*(PtCl(C_6_F_5_)(PTA)_2_) (**2a**) and its bis-cationic congeners *trans-*(PtCl(C_6_F_5_)(APTA)_2_)(BF_4_)_2_ (**2b**) and *trans-*(PtCl(C_6_F_5_)(BzPTA)_2_)(BF_4_)_2_ (**2c**). The compounds were fully characterized by multinuclear NMR, ESI-MS, elemental analysis and (for **2a**) also by single crystal X-ray diffraction, which proved the *trans* configuration of the phosphine ligands. Furthermore, in order to evaluate the cytotoxic activities of all complexes the normal human dermal fibroblast (NHDF) cell culture were used. The antineoplastic activity of the investigated compounds was checked against the human lung carcinoma (A549), epithelioid cervix carcinoma (HeLa) and breast adenocarcinoma (MCF-7) cell cultures. Interactions between the complexes and human serum albumin (HSA) using fluorescence spectroscopy and circular dichroism spectroscopy (CD) were also investigated.

## 1. Introduction

The water-solubility of 1,3,5-triaza-7-phosphaadamantane (PTA), its diverse mode of coordination, ease of functionalization and modification have made this compound a ligand of choice in coordination chemistry [1,2,3], catalysis [4,5,6,7,8,9,10,11,12] and bioinorganic chemistry [13,14]. N-Functionalization is the simplest modification of PTA; it leaves the cage structure and the P-coordinating ability intact and affords a series of cationic species with tunable properties in terms of hydrophilicity and reactivity. For these reasons, this scope has been addressed in the last decade by synthesizing derivatives bearing alkyl (methyl [15,16,17], ethyl and propyl [18,19,20], butyl [21] or hexadecyl [22]), benzyl [23,24,25] or pyridyl [26] groups.

Since the development of new metallodrugs for cancer treatment based on tertiary phosphines/aminophosphines [27,28,29,30,31,32] the interest for these types of ligands has reemerged. Both PTA and its derivatives have been applied in the synthesis of biologically active Pt(II/IV) [33,34,35,36,37], Au(I) [38,39,40], Ru(II) [41,42,43,44,45,46,47], Ag(I) [48,49,50,51] and Cu(I) [52,53,54,55] metal complexes. Indeed, phosphines can provide beneficial properties such as better stability in biological media (thus reducing undesired side reactions) and moldable affinity for polar and apolar environments in the cell (modifying the compounds transport across the membrane and their accumulation). Moreover, the choice of cationic N-functionalized PTA ligands can in principle increase the interaction with the polyanionic DNA strands [56].

An approach to circumvent cross-resistance is to change the functionality on the metal center namely by decreasing the number of available coordination sites. Monofunctional complexes can be active [57,58,59,60] and have the advantage of having ancillary ligands, which, as such, modify sterically and electronically the behavior of the resulting complex.

Recently Moreno et al. have successfully used the pentafluorophenyl group as a stabilizing ancillary ligand toward platinum for biological applications [61,62]. The C_6_F_5_ ligand has higher *trans*-effect than amines in labilizing the Pt–Cl bond [63] and the ability to increase the Lewis acidity of platinum [64,65,66,67] while modifying the hydro/lipophilicity of the complex.

Spectroscopic studies involving transition metal complexes and blood proteins such as the human serum albumin (HSA) are essential for the understanding of the biological activity of the drugs in terms of nature and strength of the interactions [68,69]. Protein–drug interactions are responsible for the distribution of the drug in the body and affect the chemotherapy effectivity and toxicity [70]. The standard example, cisplatin reaches 50–61% human serum albumin fixation after intravenous administration [71,72]. This binding is considered to be practically irreversible because only less than 5% of the drug is released from the system [73].

Platinum(II) complexes with *trans* geometry have proved to be interesting candidates to replace the commonly used cisplatin in view of its known adverse effects (mainly its toxicity) and to overcome acquired resistance (due to DME, drug-metabolizing enzymes or genetic variation) [74,75,76,77]. Trans-bisphosphino Pt(II) complexes have shown antiproliferative activity, being able to generate reactive oxygen species, targeting both mitochondria and genomic DNA [78], but also showed affinity toward protein models comparable or higher than cisplatin [79]. 

In our attempt to contribute to the knowledge of phosphine *trans*-platinum compounds and their biological activity, we synthesized the PTA (**1a**) N-allyl and N-benzyl tetrafuoroborate derivatives 1-allyl-1-azonia-3,5-diaza-7-phosphaadamantane (APTA(BF_4_), **1b**) and 1-benzyl-1-azonia-3,5-diaza-7-phosphaadamantane (BzPTA(BF_4_), **1c**) and the water soluble Pt(II) hexafluorophenyl complexes *trans-*(PtCl(C_6_F_5_)(PTA)_2_) (**2a**), *trans-*(PtCl(C_6_F_5_)(APTA)_2_)(BF_4_)_2_ (**2b**) and *trans-*(PtCl(C_6_F_5_)(BzPTA)_2_)(BF_4_)_2_ (**2c**; Figure 1).

All the new compounds have been fully characterized and the Pt(II) complexes were evaluated with NHDF (normal human dermal fibroblasts), A549 (human lung carcinoma), HeLa (human epithelioid cervix carcinoma) and MCF7 (human breast adenocarcinoma) cell cultures. NHDF cell line was used only as a control. Cancer cell lines were chosen based on cisplatin utilization in their standard treatment protocols. In addition, the interactions between the platinum complexes and HSA were investigated by circular dichroism (CD) and fluorescence spectroscopy. The association constants, acting forces and protein’s secondary structure changes are discussed based on the obtained results.

## 2. Experimental

### 2.1. General 

^1^H, ^13^C{^1^H}, ^31^P{^1^H} and ^19^F NMR spectra at 298 K were run on a Bruker AC200 spectrometer operating at 200.13, 50.323, 81.015 and 188.25 MHz, respectively; *δ* values (ppm) are reported relative to SiMe_4_, 85% H_3_PO_4_, and CFCl_3_ as reference. ESI-MS analyses were performed using LCQ-Duo (Thermo-Finnigan) operating in the positive ion mode. Instrumental parameters: capillary voltage 10 V, spray voltage 4.5 kV; capillary temperature 200 °C; mass scan ranged from 150 to 2000 m/z; N_2_ was used as sheath gas and the He pressure inside the trap was kept constant. The pressure directly read by an ion gauge (in the absence of the N_2_ stream) was 1.33 × 10^−5^ Torr. Sample solutions were prepared by dissolving the compounds in acetonitrile. Sample solutions were directly infused into the ESI source by a syringe pump at 8 μL/min flow rate. Elemental analyses were performed by the Microanalysis Laboratory of the Department of Chemical Sciences of the University of Padova.

### 2.2. Materials and Methods 

All synthetic steps were carried out under dinitrogen atmosphere using standard Schlenck techniques. Solvents were dried and purified according to standard methods [80]. 

(Pt(C_6_F_5_)(*μ*-Cl)(tht))_2_ [81,82], PTA [15] and BzPTA(Cl) [23] were synthesized following procedures reported in the literature. 

### 2.3. Cell Cultures

NHDF (normal human dermal fibroblasts; PromoCell, C-12302), A549 (human lung carcinoma; ATCC, No CCL-185 ^TM^), HeLa (human cervix carcinoma; ATCC, No CCL-2 ^TM^) and MCF7 (human breast adenocarcinoma; ATCC, No HTB-22 ^TM^) were used. NHDF, A549, HeLa and MCF7 cell lines were cultured in DMEM (Lonza, Basel, Switzerland). Media were supplemented with 10% FBS (Biological Industries, Kibbutz Beit-Haemek, Israel), 4 mM L-glutamine (Biological Industries, Kibbutz Beit-Haemek, Israel), 100 U/mL of penicillin and 100 µg/mL of streptomycin (Sigma, Steinheim, North Rhine-Westphalia, Germany).

### 2.4. Cell Viability Assays

All cell cultures were inserted in a 96-well plate (Eppendorf, Freie und Hansestadt Hamburg, Germany) in a concentration of 10^5^ cells per well. 

Compounds **2a–c** and ligands **1a–c** were dissolved in distilled water and diluted to afford concentrations of 30, 3, 0.3 and 0.03 μg/mL in DMEM and incubated in standard conditions for 72 h, after which the MTT (Sigma, Steinheim, North Rhine-Westphalia, Germany) assay was carried out. 20 µL of MTT (5 mg/mL) was added to each well, incubation for 4 h at 37 °C was implemented, and then 80 µL of lysis buffer was added. The test consists in the enzymatic reduction of the tetrazolium salt MTT in metabolically active cells. The metabolite, purple-colored formazan is measured colorimetrically, using a multiwell plate reader. The optical density (OD) was measured using a spectrophotometric microplate reader (Multiscan Go, Thermo Fisher, Waltham, MA, USA) at 570 nm with reference wavelength of 630 nm. The viability of the investigated cell cultures was estimated using following formula: viability% = (average OD for test group/average OD for control group) × 100. The untreated cells were used as a control group.

### 2.5. Cytotoxic Assay-Quantitative Suspension Test According to EN 14476

NHDF, A549, HeLa and MCF7 cells at a density of 4 × 10^4^ cells/mL were incubated in 96-well polystyrene plate (NUNC, Roskilde, Zealand, Denmark) for 24 h. Substances were tested using EN 14476 [83]. Product test solutions were prepared in 10% DMSO and DMEM supplemented with additional 10% FBS and L-glutamine. Solutions of the complexes **2a–c** and pro-ligands at concentrations from 300 μg/mL to 3 × 10^−9^ μg/mL were prepared and transferred (100 μL) into cell culture units containing 100 μL of cells suspension. Eight units were inoculated with each dilution. Plates were incubated in 37 °C/5% CO_2_ and observed daily for up to 4 days for the development of cytotoxic effects, using an inverted microscope (Olympus Corp., Hamburg, Germany; Axio Observer, Carl Zeiss, MicroImaging GmbH, Baden-Württemberg, Germany).

### 2.6. Octanol–Water Partition Coefficient Determination

The log(P) values corresponding to the octanol–water partition coefficient was adjusted to the solubility properties of the compounds [84]. Complexes were dissolved in water previously saturated with octanol, to achieve concentrations of 10^−3^ M. At 24 °C, 25 mL of octanol saturated with water was introduced into a 100 mL flask equipped and then 25 mL of the aqueous complex solutions was added. The two-phase mixture thus obtained was stirred vigorously for 20 min. After separation of the phases and removal of the respective solvents under vacuum, the attained residues were weighed. Values for log(P) of −0.11, −0.71 and −0.40 for **2a**, **2b** and **2c**, respectively, were obtained.

### 2.7. HSA Interaction

High purity free-HSA (>96%, Sigma-Aldrich, Steinheim, North Rhine-Westphalia, Germany) was used as purchased. The stock solutions of platinum complexes (200 μM) were freshly prepared in double distilled water prior to their use. The HSA solutions (200 μM) were prepared in PBS (pH 7.40). In the final step, the samples contained (protein):(drug) equal 1:0–1:9 in PBS, where C_HSA_ was 10 μM. Measurements were done after 24 h incubation at 300, 305 and 310 K. HSA concentration was determined, using A_279_ (1 mg/mL) as 0.531 [85]. 

#### 2.7.1. Fluorescence Measurements

Emission fluorescence spectra were recorded using Hitachi F-2700 spectrofluorimeter (Tokio, Japan) in 1.0 cm quartz cells in the 300–400 nm range. Trp-214 fluorescence of HSA was measured at 295 nm. The intensity at 335 nm (Trp-214) was used to achieve the quenching mechanism and calculate the binding constants and thermodynamic parameters according to literature reports.

#### 2.7.2. CD Measurement

Circular dichroism spectra were recorded on a Jasco J-715 spectropolarimeter, over the range of 190–250 nm in 0.1 cm cuvettes. The α-helical content of HSA was calculated from the molar ellipticity (*θ*) at 208–210 nm using Equations (1) and (2) [86]:
(1)$$MRE_{209}=\frac{Observed_{CD}\;\left(mdeg\right)}{C_{p}\cdot n\cdot l\cdot 10},$$
(2)$$\alpha -helix\;\left(\%\right)=\frac{MRE_{209}-4000}{33000-4000}\times 100,$$
where *MRE* is the mean residue ellipticity, *C**_p_* is the molar concentration of the protein, *n* is the number of amino acid residues (583) and *l* is the path-length (0.1 cm).

### 2.8. Synthesis and Analytical Data

APTA(BF_4_) **1b.** Of PTA (6.36 mmol) 1.00 g was dissolved in 50 mL of anhydrous acetone at room temperature and treated under stirring with 0.59 mL of allyl iodide (1.08 g, 6.45 mmol). After 1 h reflux, the volume of the solution was taken to ca. 10 mL and added with 15 mL of diethyl ether to give a white precipitate. The solid was filtered, dried under vacuum and dissolved in 50 mL of anhydrous methanol. After adding 1.88 g of TlBF_4_ (6.45 mmol), the clear solution was left stirring at room temperature for 30 min. The precipitate of TlI was filtered off and the solution was concentrated on a rotary evaporator. The product was obtained by precipitation with diethyl ether, subsequently washed three times with 5 mL of Et_2_O, recrystallized from methanol, and dried under vacuum. Yield 1.51 g, 83.3%. Anal. Calc. for C_9_H_17_BF_4_N_3_P: C, 37.92; H, 6.01; N, 14.74%; Found: C, 37.77; H, 5.98; N, 14.68%. ^1^H NMR (*δ*, D_2_O): 5.99–5.78 (m, 1H, C*H*CH_2_N^+^); 5.68–5.54 (m, 2H, C*H*_2_CH); 4.80 (dd,^1^*J*_HH_ = 12 Hz, 4H, N^+^C*H*_2_N); 4.41 (dd, ^1^*J*_HH_ = 14 Hz, 2H, NC*H*_2_N); 4.20 (d, ^2^*J*_PH_ = 14 Hz, 2H, N^+^C*H*_2_P); 3.87–3.66 (m, 4H, NC*H*_2_P); 3.47 (d, ^1^*J*_HH_ = 8 Hz, 2H, CHC*H*_2_N^+^); ^13^C{^1^H} NMR (*δ*, D_2_O): 130.5 (*C*HCH_2_N^+^); 122.5 (*C*H_2_CH); 79.1 (N^+^*C*H_2_N); 70.0 (NCH_2_N); 65.7 (CH*C*H_2_N^+^); 53.7 (d, ^1^*J*_PC_ = 33 Hz, N^+^*C*H_2_P); 46.2 (d, ^1^*J*_PC_ = 21 Hz, N*C*H_2_P); ^31^P{^1^H} NMR (*δ*, D_2_O): −82.8 (s); ^19^F{^1^H} NMR (*δ*, D_2_O): −150.4 (BF_4_). ESI-MS (CH_3_OH): *m/z* 198.05 (100%) [M − BF_4_]^+^.

BzPTA(BF_4_) **1c.** Of BzPTA(Cl) (0.70 mmol) 0.20 g was dissolved in 50 mL of anhydrous methanol at room temperature and added with 0.20 g (0.70 mmol) of TlBF_4_. After stirring for 1 h the precipitate of TlCl was filtered off and the solution was concentrated on a rotary evaporator. The solid product obtained by precipitation with diethyl ether was washed three times with 5 mL of Et_2_O, recrystallized from methanol, and dried under vacuum. Yield 0.24 g, 96.0%. Anal. Calc. for C_13_H_19_BF_4_N_3_P: C, 46.60; H, 5.72; N, 12.54%; Found: C, 46.87; H, 5.68; N, 12.18%. ^1^H NMR (*δ*, D_2_O): 7.50–7.40 (m, 5H, Ar); 4.79 (dd, ^1^*J*_HH_ = 14 Hz, 4H, N^+^C*H*_2_N); 4.33 (dd, 2H, ^1^*J*_HH_ = 14 Hz, NC*H*_2_N); 4.16 (d, ^2^*J*_PH_ = 6 Hz, 2H, N^+^C*H*_2_P); 4.07 (s, 2H, PhC*H*_2_N^+^); 3.92–3.59 (m, 4H, NC*H*_2_P); ^13^C{^1^H} NMR (*δ*, D_2_O): 132.9, 130.9, 129.3, 124.5 (Ph); 78.7 (N^+^*C*H_2_N); 69.4 (NCH_2_N); 66.8 (Ph*C*H_2_N^+^); 52.8 (d, ^1^*J*_PC_ = 33 Hz, N^+^*C*H_2_P); 45.6 (d, ^1^*J*_PC_ = 21 Hz, N*C*H_2_P); ^31^P{^1^H} NMR (*δ*, D_2_O): −82.85 (s); ^19^F{^1^H} NMR (*δ*, D_2_O): −150.4 (BF_4_). ESI-MS (CH_3_OH): *m/z* 248.06 (100%) [M − BF_4_]^+^.

(PtCl(C_6_F_5_)(PTA)_2_) **2a.** Of (Pt(*μ*-Cl)(C_6_F_5_)(tht))_2_ (0.21 mmol) 0.20 g were dissolved in 30 mL of anhydrous dichloromethane at room temperature and treated with 0.13 g of PTA (0.83 mmol). After stirring for 1 h, the solution was concentrated on rotary evaporator and treated with diethyl ether. The solid precipitated as a white powder was filtered on a Gooch funnel, washed three times with 5 mL of Et_2_O, and dried under vacuum. **2a** is soluble in DMSO and dichloromethane, sparingly soluble in H_2_O (S_25 °C_ ≈ 0.35 mg mL^−1^), MeOH and EtOH and insoluble in diethyl ether, C_6_H_6_ and alkanes. Yield 0.28 g, 99%. Anal. Calc. for C_18_H_24_ClF_5_N_6_P_2_Pt: C, 30.37; H, 3.40; N, 11.81%; Found: C, 30.58; H, 5.48; N, 11.61%. ^1^H NMR (*δ*, CD_2_Cl_2_): 4.47 (m, 12H, PC*H*_2_N); 4.09 (s, 12H, NC*H*_2_N); ^13^C{^1^H} NMR (*δ*, CD_2_Cl_2_): 73.1 (s, NCH_2_N); 49.7 (m, N*C*H_2_P); ^31^P{^1^H} NMR (*δ*, CD_2_Cl_2_): −61.8 (s, ^1^*J*_Pt-P_ = 2453 Hz). ^19^F{^1^H} NMR (*δ*, CD_2_Cl_2_): −118.3 (m, *o*-F, ^3^*J*_Pt-F_ = 435 Hz, ^3^*J*_F-F_ = 14.1 Hz), −160.5 (m, *p*-F, ^3^*J*_F-F_ = 14.1 Hz), −162.3 (m, m-F). ESI-MS (CH_3_OH): *m/z* 713.08 (100%) [M + H]^+^. Crystals suitable for X-Ray analysis were obtained by slow growth through diffusion of Et_2_O in a solution of **2a** in CH_2_Cl_2_.

(PtCl(C_6_F_5_)(APTA)_2_)(BF_4_)_2_
**2b.** Through a synthetic procedure similar to the one described for complex **2a**, complex **2b** was obtained from 0.05 g of (Pt(*μ*-Cl)(C_6_F_5_)(tht))_2_ (0.05 mmol) dissolved in 10 mL of methanol and treated with 0.06 g of APTA(BF_4_) (0.21 mmol). **2b** is soluble in DMSO, dichloromethane and H_2_O (S_25°C_ ≈ 4.0 mg mL^−1^), sparingly soluble in MeOH and EtOH and insoluble in diethyl ether, C_6_H_6_ and alkanes. Yield 0.078 g, 79%. Anal. Calc. for C_24_H_34_B_2_ClF_13_N_6_P_2_Pt: C, 29.79; H, 3.54; N, 8.69%; Found: C, 30.03; H, 3.68; N, 8.21%. ^1^H NMR (*δ*, DMSO-d_6_): 6.04–5.87 (m, 2H, C*H*CH_2_N^+^); 5.71–5.54 (m, 4H, C*H*_2_CH); 4.99 (dd, 8H, N^+^C*H*_2_N); 4.43 (dd, 4H, NC*H*_2_N); 4.36 (s, 4H, N^+^C*H*_2_P); 4.02 (dd, 8H, NC*H*_2_P); 3.68 (d, ^3^*J*_HH_ = 6 Hz, 4H, CHC*H*_2_N^+^); ^13^C{^1^H} NMR (*δ*, DMSO-d_6_): 128.9 (*C*HCH_2_N^+^); 124.2 (*C*H_2_CH); 78.8 (N^+^*C*H_2_N); 68.7 (NCH_2_N); 63.8 (CH*C*H_2_N^+^); 49.75 (m, N^+^*C*H_2_P); 45.3 (m, N*C*H_2_P); ^31^P{^1^H} NMR (*δ*, DMSO-d_6_): −41.42 (s, ^1^*J*_Pt-P_ = 2628 Hz). ^19^F{^1^H} NMR (*δ*, DMSO-d_6_): −117.2 (m, *o*-F, ^3^*J*_Pt-F_ = 395 Hz, ^3^*J*_F-F_ = 22.6), −148.1 (BF_4_), −159.5 (t, *p*-F, ^3^*J*_F-F_ = 22.6 Hz), −161.1 (m, *m*-F). ESI-MS (CH_3_OH): *m/z* 880.05 (100%) [M − BF_4_]^+^.

(PtCl(C_6_F_5_)(BzPTA)_2_)(BF_4_)_2_
**2c.** Through a synthetic procedure similar to the one described for complex **2a**, complex **2c** was obtained from 0.05 g of (Pt(*μ*-Cl)(C_6_F_5_)(tht))_2_ (0.05 mmol) in 10 mL of methanol and 0.07 g of BzPTA(BF_4_) (0.21 mmol). **2c** is soluble in DMSO, dichloromethane and H_2_O (S_25°C_ ≈ 3.0 mg mL^−1^), sparingly soluble in MeOH and EtOH and insoluble in diethyl ether, C_6_H_6_ and alkanes. Yield 0.085 g, 77%. Anal. Calc. for C_32_H_38_B_2_ClF_13_N_6_P_2_Pt: C, 36.00; H, 3.59; N, 7.87%; Found: C, 36.12; H, 3.51; N, 7.80%. ^1^H NMR (*δ*, DMSO-d_6_): 7.60–7.36 (m, 10H, Ar); 5.03 (dd, 8H, N^+^C*H*_2_N); 4.44 (dd, 4H, NC*H*_2_N); 4.22 (s, 4H, PhC*H*_2_N^+^); 4.15 (s, 4H, N^+^C*H*_2_P); 4.03 (m, 8H, NC*H*_2_P); ^13^C{^1^H} NMR (*δ*, DMSO-d_6_): 133.3, 131.0, 129.5, 125.7 (Ph); 79.0 (N^+^*C*H_2_N); 68.9 (NCH_2_N); 64.7 (Ph*C*H_2_N^+^); 49.1 (m, N^+^*C*H_2_P); 45.3 (m, N*C*H_2_P); ^31^P{^1^H} NMR (*δ*, DMSO-d_6_): −40.8 (s, ^1^*J*_Pt-P_ = 2627 Hz). ^19^F{^1^H} NMR (*δ*, DMSO-d_6_): −117.3 (m, *o*-F, ^3^*J*_Pt-F_ = 393 Hz, ^3^*J*_F-F_ = 22.6), −148.2 (BF_4_), −159.5 (t, *p*-F, ^3^*J*_F-F_ = 22.6 Hz), −161.1 (m, *m*-F). ESI-MS (CH_3_OH): *m/z* 980.10 (100%) [M − BF_4_]^+^.

### 2.9. X-ray Crystallography

An X-ray quality crystal of **2a** was immersed in cryo-oil, mounted in a Nylon loop and measured at 153 K. Intensity data was collected using a Bruker AXS-KAPPA APEX II diffractometer with graphite monochromatic Mo-K*α* (*λ* = 0.71073) radiation. Data were collected using phi and omega scans of 0.5° per frame and a full sphere of data were obtained. Cell parameters were retrieved using Bruker SMART software and refined using Bruker SAINT [87] on all the observed reflections. Absorption corrections were applied using SADABS [88]. Structures were solved by direct methods by using SIR97 [89] and refined with SHELXL-2014/7 [90]. Calculations were performed using the WinGX System (Version 2014.1) [91]. The hydrogen atoms attached to the methylene carbons were inserted in calculated positions, their Uiso (H) defined as 1.2 Ueq of the parent carbon atoms. Least square refinements with anisotropic thermal motion parameters for all the non-hydrogen atoms and isotropic for most of the remaining atoms were employed. The molecular structure of **2a** in represented in Figure 1, crystallographic details are listed in Table 1 and selected bond distances and angles in the legend of Figure 1. CCDC 1572427 contains the supplementary crystallographic data for this paper. These data can be obtained free of charge from The Cambridge Crystallographic Data Centre via www.ccdc.cam.ac.uk/data_request/cif.

Crystal data for **2a**: C_18_H_24_ClF_5_N_6_P_2_Pt, *M* = 711.91, *T* = 153(2) K, monoclinic, space group *C*2/c, *a* = 19.9537(17) Å, *b* = 12.2786(9) Å, *c* = 10.2981(9) Å, *β* = 115.314(5)°, *V* = 2280.8(3) Å^3^, *Z* = 4, *D*c = 2.073 g/cm^3^, *μ* = 6.471 mm^−1^, 10377 reflections collected, 1928 unique, *I > 2σ(I)* (*R_in_*_t_ = 0.0383), *R_1_* = 0.0183, *wR2* = 0.0427, GOF 1.043. CCDC 1572427.

## 3. Results and Discussion

### 3.1. Synthesis and Characterization

1,3,5-triaza-7-phosphaadamantane (PTA, **1a**) was allowed to react with allyl iodide and benzyl chloride [8a] in acetone under reflux to give the iodide salt of 1-allyl-1-azonia-3,5-diaza-7-phosphaadamantane (APTA(I)) and benzyl derivative 1-benzyl-1-azonia-3,5-diaza-7-phosphaadamantane (BzPTA(Cl)) in high yields, respectively. The exchange of the halide counterions by tetrafluoroborate on both the latter derivatives afforded the pro-ligands APTA(BF_4_) (**1b**) and BzPTA(BF_4_) (**1c**) as white and air-stable solids (Scheme 1). The presence of the BF_4_ anion proved to increase the stability to the N-alkylated PTAs and its non-coordinating property was fundamental for the preparation of the platinum(II) complexes.

^1^H NMR spectra of **1b** and **1c** (see the Experimental Section) show the typical signals of N-monosubstituted PTA: (i) two AB resonances around 4.8 and 4.4–4.3 ppm for the protons in the lower rim revealing their diastereotopic nature; (ii) a doublet at 4.2 ppm and a ABX-type multiplet at 3.5–3.9 ppm for the protons in the upper rim, enlightening the coupling to phosphorus and (iii) the expected signals for the allyl and benzyl group [4c]. The ^13^C{^1^H} NMR spectra are characterized by the expected six singlets of the unsymmetrical alkylated PTA cage in the range 45–70 ppm and, in both cases, the singlet due to the methylene group at around 79 ppm. The signals of the sp^2^ C in the allyl group (2 peaks) of **1b** and the benzylic phenyl ring (5 peaks) of **1c** are in the range 122–133 ppm. The signal for the substituted phenyl C has not been detected.

The ^31^P{^1^H} NMR spectra show a single peak at about −83 ppm for both derivatives, which is upfield from the resonance reported for PTA (−98.5 ppm) [92] and very similar to those reported for other N-substituted PTA compounds [15,16,17,18,19,20,21,22,23,24,25,26].

The platinum pentafluorophenyl complexes *trans-*(PtCl(C_6_F_5_)(PTA)_2_) (**2a**) and the bis-cationic *trans-*(PtCl(C_6_F_5_)(APTA)_2_)(BF_4_)_2_ (**2b**) and *trans-*(PtCl(C_6_F_5_)(BzPTA)_2_)(BF_4_)_2_ (**2c**) have been obtained as white powders with yields in the range 77–99% by reaction of 4 equivalents of **1a**, **1b** or **1c** with the binuclear complex *trans-*(Pt(μ-Cl)(C_6_F_5_)(tht))_2_ (tht = tetrahydrothiophene; Scheme 2).

Complex **2a** is sparingly soluble in water but well soluble in DMSO and other polar solvents and in medium polarity solvents such as dichloromethane. Compounds **2b–c** show a similar behavior but with higher solubility in water due to their cationic nature.

The novel compounds **2a–c** were characterized by elemental analysis, as well as by ^1^H (Appendix A), ^31^P{^1^H}, ^13^C{^1^H} and ^19^F{^1^H} NMR spectroscopies (see the Experimental Section). The ^1^H NMR spectra confirm the presence of the phosphines: their resonances are shifted downfield by the electron withdrawing influence of the metal center, and the ^2^*J*_PH_ coupling constants present lower values. The ^31^P{^1^H} NMR spectra show a singlet, as expected for equivalent nuclei in a *trans* geometry. The presence of the phosphine in the Pt(II) complexes is confirmed by the downfield shift, Δ*δ*, of the ^31^P resonance upon coordination: 36.7 (**2a**), 41.4 (**2b**) and 42.1 ppm (**2c**). The P-Pt coupling constants are of 2450 Hz for **2a** and ca. 2628 Hz for **2b** and **2c**. In complex **2a**’s ^13^C{^1^H} NMR spectrum, the signals for the upper rim (49.67 ppm) and for the lower rim (73.11 ppm) carbons can be seen, while those of complexes **2b–c** show the typical structure for alkylated PTA cages with peaks in the range 45–69 ppm. In both cases the singlet of the methylene group is around 79 ppm, as for the pro-ligands (see above). Similarly, the signals of the olefinic carbons (2 peaks) in the APTA^+^ moiety of **2b** and the benzylic phenyl ring (5 peaks) of BzPTA^+^ in **2c** are in the range 124–135 ppm. No resonances for the pentafluorophenyl carbons have been detected. However, the presence of this group was confirmed by the ^19^F{^1^H} NMR spectra where the ^3^*J*_Pt-F_ coupling of the *o*-F nuclei with the ^195^Pt metal center assume values in the range 440–390 Hz. The ESI-MS spectra of the pro-ligands and the complexes were performed in methanol and confirm the expected formulations.

X-ray quality crystals of complex **2a** (Figure 2) have been obtained by slow diffusion of diethyl ether on a dichloromethane solution of the compound in air and at room temperature. It crystalized in the monoclinic space group *C*2/c, its asymmetric unit including a PTA phosphine ligand and half of the fluorinated moiety. Resulting from the two-fold axis that runs across the molecule, a mononuclear compound is thus constructed from a Pt(II) cation, which is bound to a chloride anion, P-bound to two PTA phosphines and C-bound to one pentafluorobenzene anion, therefore giving rise to a square planar geometry (*τ*_4_ = 0.03) [93]. The pentafluorophenyl ring lies perpendicular to the plan constructed by the Pt-, P- and Cl-atoms (89.8°). The bond lengths involving the metal and the donor atoms (see Figure 1, legend), are within those found in other Pt(II) compounds [94,95,96,97,98,99]. No classical H-bond interactions could be found in **2a**, but the shortest C–H⋅⋅⋅N interactions (Figure 1) expand the structure to the third dimension.

### 3.2. Biologic Assays

Cancer cell lines used in this study belong to one type of neoplasms, derived from epithelial. Those carcinomas hardly respond for treatment and have high potential for metastasis. The fibroblast cell line reflects a normal cell of body. Cytotoxic Assay-Quantitative Suspension Test According to EN 14476 was done as a preliminary investigation. The results were similar to those obtained in MTT test. Considering that MTT method is more detailed, analysis and discussion were carried out on the basis of these results. The IC50 (half maximal inhibitory concentration) values, after 72 h of incubation with the tested compounds, are shown in Table 2.

From Table 2, we can see that all three complexes show cytotoxicity against the fibroblasts cell line (NHDF). Compound **2a** and **2b** shows higher cytotoxicity to A549 and lower to HeLa and MCF7 cell lines, while compound **2c** proved more active against HeLa cell lines. All three complexes show cytotoxicity against fibroblasts cell line (NHDF), but lower when compared to A549 or HeLa cell lines. The lower cytotoxicity effect of cisplatin on A549 cell line can be caused by different growing conditions, medium supplementation and exposition time. Cisplatin displayed, in contrast to tested compounds, weaker activity against the chosen neoplastic cell lines (especially A549); comparable doses of cisplatin are toxic for both normal and neoplastic (HeLa) cell lines. It is also worth mentioning that the solubility and stability of compounds **2a–c** in aqua-media were higher than those of cisplatin. Hydrolytic stability tests of **2a–c**, performed at 2.0 mM concentration in NaCl solution (5 or 100 mM in D_2_O-d_6_/DMSO 10/1) by ^1^H and ^31^P NMR spectroscopy, shows no free phosphine after 72 h at room temperature and only partial hydrolysis of the Pt–Cl bond. The bioactivities of **2a–c** coincided with values of their logarithm of 1-octanol/water partition coefficient (log(P), see Experimental). This method is one of the most widely used ones to describe hydrophobic/hydrophilic properties of chemicals. For the accurate biological activity and bioavailability of potential drugs, a balanced solubility in both water and nonpolar compounds such as lipids is required [54]. Indeed, the different activity of the platinum complexes **2a–c** could be related to their more balanced log(P) values, in contrast to cisplatin with strongly negative log(P) factor (−2.21) [101]. The lower cytotoxicity of the new compounds for normal cells in comparison to cancer cells was their undoubted advantage. Even if their cytotoxicity for cancer cells was not radically lower than cisplatin, they could be a reasonable alternative, especially useful in cisplatin resistant therapies. Their good solubility in aqueous solutions is also important, which will allow us to “relieve” therapy with toxic solvents. Further research is thus needed to proof usefulness of the tested compounds in tumor therapy.

### 3.3. HSA Interactions

The effect of compounds **2b** and **2c** on the fluorescence intensity of HSA (max 335 nm) is shown in Figure 3. It decreased with increasing amounts of the complexes, indicating that some interactions between the two components were taking place. The variation in intensity results from a quenching effect of the metal compounds. In addition, a maximum blue shift of ca. 3 (with **2b**) and 5 (with **2c**) nm was observed, indicating that Trp-214 was placed in a more hydrophobic environment and was less exposed to the solvent. It also suggests that the interaction might occur via the hydrophobic region of the protein [102,103].

The Stern−Volmer plots of HSA in the presence of the platinum compounds as quenchers, at different conditions, are shown in Appendix A. The calculated Stern-Volmer constants (K_SV_) and quenching rate constants (Kq) for the interactions between HSA and complexes **2b** and **2c** are listed in Table 3. The obtained results showed that Ksv increased with the increase in temperature, thus indicating that the fluorescence quenching mechanism of HSA by the Pt compounds was a dynamic one. The maximum expected value of Kq for dynamic processes is 2 × 10^10^ M^–1^·s^–1^ [103,104], but the higher value obtained in our study, ca. 10^12^ M^–1^·s^–1^, shows that specific drug–protein interactions were involved, making Kq larger [105]. It suggests that the fluorescence quenching processes of the HSA-Pt complex systems were initiated by a combined process of dynamic and static quenching.

Fluorescence quenching data were also analyzed to obtain various binding parameters for the interaction of the platinum complexes with HSA, as shown in Appendix A. The association constants (K_A_) for complexes **2b** and **2c** and the number of binding site (*n*) were calculated using Appendix A and are listed in Table 3. The results show that the association constants K_A_ increased with the temperature, indicating the formation of stable adducts and endothermic processes; in addition, the values of *n* approximated to 1 suggest only one reactive site for both complexes in the protein. The interactions of HSA with **2b** and **2c** were relatively stronger than that with cisplatin; under the same conditions the binding constant Ka for cisplatin-HSA complex was of 8.52 × 10^2^ M^−1^ [106]. Nevertheless, they were comparable to those obtained with ruthenium compounds and other platinum complexes (10^4^–10^5^ M^−1^) [72,107,108,109,110]. 

Intermolecular interactions between a drug and a protein may involve van der Waals forces, hydrogen bonds, electrostatic forces and hydrophobic interactions [111]. To clarify the type of reaction between our complexes and HSA the thermodynamic parameters were calculated using the van’t Hoff plots and Appendix A. Table 4 shows the values of ΔH^0^ and ΔS^0^ obtained from such relationships as well as the corresponding values for Gibbs free energy change (ΔG^0^). The positive ΔH^0^ and ΔS^0^ for complexes **2b** and **2c** suggest that hydrophobic and ionic interactions might be involved. The negative values of ∆G^0^ indicate that the interactions were spontaneous. The aromatic ring present in both complexes **2b** and **2c** enabled hydrophobic and π–π stacking interactions with amino acid residues in HSA such as Trp, Tyr and His; electrostatic interactions were also reasonable. It is known that cisplatin interacts with HSA generating irreversible covalent bonds but our compounds form reversible adducts via noncovalent interactions.

Circular dichroism (CD) experiments were carried out to verify the binding of Pt(II) complexes to HSA and their effect on the protein secondary structure. The observed negative bands at wavelength of 208–209 and 222–223 nm (Appendix A) were characteristic of an α-helical structure of proteins. The binding of **2b** and **2c** to HSA reduced both bands. The *θ* values of the protein decreased with the increase in concentration of the platinum complexes with shifts in the bands positions indicating some loss of α-helical secondary structure. According to Equation (2) (see Experimental), the content of *α*-helix decreased from 50.88% to 46.29% and to 47.17% at an HSA/complex molar ratio of 1/10 for **2b** and **2c**, respectively. These changes indicate an interaction between our compounds and amino acid residues of the protein, which partially destroy existing hydrogen bonds. Nevertheless, the absence of changes in the shape of the spectra suggests the protein’s secondary structure is still dominated by α-helix forms.

## 4. Conclusions

The new Pt(II) pentafluorophenyl complex (PtCl(C_6_F_5_)(PTA)_2_) (**2a**) and its bis-cationic congeners with allyl-PTA(BF_4_) (**2b**) and benzyl-PTA(BF_4_) (**2c**) derivatives were synthesized and fully characterized by multinuclear NMR, ESI mass spectrometry and elemental analysis. All the complexes had a *trans* configuration of the aminophosphine ligands as confirmed by the PTA derivative X-ray structure obtained for the PTA derivative. Their biological activity was tested against NHDF, A549, HeLa and MCF7. The results have shown that **2a** and **2b** compounds had strong cytotoxicity against A549 and **2c** against HeLa cell lines. All three complexes showed lower cytotoxicity against fibroblasts cell line (NHDF). Cisplatin displayed the weakest cytotoxic activity against the chosen neoplastic cell lines (especially A549) in comparison to tested compounds. In addition, our compounds provided the great advantage of water solubility making this a positive feature against the frequently used DMSO. Complexes **2b** and **2c** interacted with human serum albumin causing a conformational change with the loss of helical stability of the protein. The binding affinity to HSA of the benzyl derivative **2c** was higher than that of the allyl **2a**. Concerning the thermodynamic parameters, the positive values of ∆H^0^ and ∆S^0^ and negative ∆G^0^ for both complexes suggest that hydrophobic and ionic interactions participated in the interaction with HSA and the interaction of complexes with the protein was a spontaneous process. Further studies are necessary to assess the mechanism of action, to address the future synthetic efforts toward more active species.

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
