# Peer review of "Pentafluorophenyl Platinum(II) Complexes of PTA and Its N-Allyl and N-Benzyl Derivatives: Synthesis, Characterization and Biological Activity"

_materials, 2019, doi:10.3390/ma12233907_

Round 1
Reviewer 1 Report
In this study, the authors synthesized the new 1,3,5-triaza-phosphaadamantane-based pentafluorophenyl Pt (II) complexes. Successful synthesis is an interesting result, , however the study of available to verify its usefulness is poor. Therefore, the following contents should be revised and supplemented.1. The only human dermal fibroblast cell line was used to assess toxicity. Possibility of no interference with normal lung, breast, cervical endothelial or epithelial cells should be discussed additionally.
2. There is no rationale for utilizing three cancer types (lung, cervical, breast) representatively.
3. Human serum albumin is involved in the transport of drugs, and the purification of toxicity is determined by drug-metabolizing enzymes (DME), including CYP. Substances that are easily degraded by CYP are less likely to be cytotoxic to cancer cells. A theoretical description of this part should be added and discussed in a certain part.
4. Line88-93: It is unclear what is the ultimate goal of performing research with such cancer cells and human dermal fibroblast cells.
5. Paragraphs 2.3 and 2.4 can be described in one paragraph by a common subject, "Cell cultures".
6. The biologic analysis data presented by the authors cannot determine the usefulness of these chemical products at all. Table 2 indicates cytotoxicity even in normal human fibroblasts cells. And viability results were not presented.
7. A549 is not resistant to cisplatin. It should be clearly discussed that the cytotoxic effect was lower on A549 by cisplatin in this study. Line 76: adverse side effects → adverse effects or side effects
Line 76: Gaining resistance is due to DME or genetic variation, not the adverse effect.
Author Response
Reviewer: 1
We thank the Reviewer for a positive evaluation of our work and for the very valuable suggestions aiming at its further improvement. When possible, the manuscript has been revised in light of those suggestions and the necessary amendments/corrections have been introduced. In the other cases a detailed answer to the Reviewer’s comment has been provided.
Comment 1: The only human dermal fibroblast cell line was used to assess toxicity. Possibility of no interference with normal lung, breast, cervical endothelial or epithelial cells should be discussed additionally.
Answer: The influence of the tested compounds on normal epithelial and endothelial cells will be investigated in future study. Normal human dermal fibroblasts were used only as a control in cytotoxicity assay. Similar utilization of NHDF cell line can be found in literature i.e. Yonghao Qi et al.: “Cinchonine induces apoptosis of HeLa and A549 cells through targeting TRAF6.”, J. Exp. Clin. Cancer Res. 2017; 36: 35. Other authors also used fibroblast cell lines, e.g. Carlson Alexander et al.: “Platinum(II) complexes of imidazophenanthroline‐based polypyridine ligands as potential anticancer agents: synthesis, characterization, in vitro cytotoxicity studies and a comparative ab initio, and DFT studies with cisplatin, carboplatin, and oxaliplatin”, J. Biol. Inorg. Chem., 2018, 23, 833; Dharmasivam Mahendiran et al.: “Copper complexes as prospective anticancer agents: in vitro and in vivo evaluation, selective targeting of cancer cells by DNA damage and S phase arrest”, RSC Adv., 2018, 8, 16973 as well as Smoleński et al.: “Light-Stable Polypyridine Silver(I) Complexes of 1,3,5-Triaza-7-Phosphaadamantane (PTA) and 1,3,5-Triaza-7-Phosphaadamantane-7-Sulfide (PTA=S): Significant Antiproliferative Activity of Representative Examples in Aqueous Media”, Dalton Trans. 2019, 48, 11235.
Comment 2: There is no rationale for utilizing three cancer types (lung, cervical, breast) representatively.
Answer: Cisplatin is routinely used in treatment of lung, cervical and breast cancers. Therefore, utilizing three cancers type cells cultures seemed to be reasonable. Furthermore, the use of these lines was due to the possibility of comparing our research results with those described in the literature.
Comment 3: Human serum albumin is involved in the transport of drugs, and the purification of toxicity is determined by drug-metabolizing enzymes (DME), including CYP. Substances that are easily degraded by CYP are less likely to be cytotoxic to cancer cells. A theoretical description of this part should be added and discussed in a certain part.
Answer: Drug-metabolizing enzymes, including CYPs (Cyt P450) are very important from a pharmacological point of view. The key role of these proteins is biotransformation of xenobiotics that means lipid-soluble drugs are transformed into more polar, water-soluble metabolites easily excreted from a body. However, xenobiotics biotransformation includes reactions leading to activation of prodrugs and procarcinogens. Therefore, the metabolites can be inactive or less active than parent drugs, but often may have enhanced activity or toxic effects. For example, anticancer drugs from mitomycin c group, nitrofurans, nitroimidazoles or 1-nitroacridine derivatives are active because of prodrugs bioreduction which occurs under the xypoxemic conditions inside the tumor cells. From the other side it was shown that isoenzyme CYP2D6 contribution in metabolism of anticancer drugs is insignificant, but it influences the effectiveness of breast cancer treatment by tamoxifen. About 90% of tamoxifen is metabolized by CYP3A4/5 to inactive form – N-dimethyltamoxifen and only ~10% is metabolized by CYP2D6 to anticancer active product 4-hydroxytamoxifen. Isoenzyme CYP2D6 transforms also inactive derivative N-dimethyltamoxifen to 4-hydroxy-N-dimethyltamoxifen with antiestrogen activity, which has therapeutic effect on breast cancer cells. Other metabolism pathways of the drug include CYP3A4/5 leading to formation α-hydroxytamoxifen responsible for cancer-producing side effects.
Nevertheless, drug-metabolizing enzymes and metabolism products of platinum complexes are not the topic of the article. CYPs enzymes belong to other group of proteins which structure and functions differ from those of HSA. Our goal was an analysis of the interaction between HSA and highly anticancer active platinum compounds in the context of forming bonds and association constants, that we were able to depict. In our opinion, the introduction of a theoretical description of the metabolism effect of the CYPs could produce some distraction for the readers, especially, when the metabolism effect of tested complexes will not be examined in this direction.
Comment 4: Line88-93: It is unclear what is the ultimate goal of performing research with such cancer cells and human dermal fibroblast cells.
Answer: Line 96-98: Sentence “NHDF cell line was used only as control. Cancer cell lines were chosen based on cisplatin utilization in their standard treatment protocols” was inserted.
Comment 5 Paragraphs 2.3 and 2.4 can be described in one paragraph by a common subject, "Cell cultures".
Answer: Paragraphs 2.3 and 2.4 were fused.
Comment 6: The biologic analysis data presented by the authors cannot determine the usefulness of these chemical products at all. Table 2 indicates cytotoxicity even in normal human fibroblasts cells. And viability results were not presented.?
Answer: The caption of Table 2 was changed to “IC50 values (μM) of the tested complexes (2a-c), ligands (1a-c), and cisplatin”. IC50 value represents cytotoxic potency of the tested compounds. MTT assay allows to measure viability of tested cells and usually only IC 50 values are presented in manuscripts. The submitted work applies only for the general bio-analysis of described compounds. The following tests and publication of their results may have important cognitive value for further studies and cancer treatment development.
Comment 7: A549 is not resistant to cisplatin. It should be clearly discussed that the cytotoxic effect was lower on A549 by cisplatin in this study.
Answer: Line 358-360 sentence: “Lower cytotoxicity effect of cisplatin on A549 cell line can be caused by different growing conditions such as level of bovine serum in medium etc.” was added. The FBS concentration in medium can influence cell viability, e.g. standard FBS concentration equals 10% and freezing medium must have higher percentage of FBS (20%) in order to increase viability after defreezing.
Comment 8: Line 76: adverse side effects → adverse effects or side effects. Line 76: Gaining resistance is due to DME or genetic variation, not the adverse effect.
Answer: The line 79 was changed and the sentence: “(due to DME, drug-metabolizing enzymes, or genetic variation)” was added.
Following your request, all changes in the manuscript are marked by a yellow background.
Additionally, following the Editor comment, we have partially changed some parts of the manuscript for decrease of a repetition rate (all changes are marked by a blue background).
Reviewer 2 Report
1. The significance of this work is not clearly mentioned. As authors mentioned, pentafluorophenyl's advantage of the biological applications is already reported. Even though authors want to replace the cisplatin with new compounds but results didn't show a clear advantage of them (no significant improvement of anti-cancer effects).
2. Authors mentioned that compounds 2b and 2c have a higher solubility than 2a but it is subjectively mentioned. Please provide the exact number of water solubility of compounds.
3. It is hard to say the tested compounds have lower cytotoxicity on fibroblast cells than cancer cells.
4. It is interesting that two different compounds showed anti-cancer effects on a specific cell type (2b for A549 and 2c for HeLa). Please provide more explanation of why it happened.
5. As Shen et al. reported (Pharmacol Rev. 2012, 64(3): 706–721), there are several mechanisms on how cells obtained the cisplatin-resistance. If authors explain how new compounds can avoid the cisplatin-resistance mechanism, it would be great to emphasize their contribution not only to pharmaceutical chemistry but also cancer therapy.
Author Response
We thank the Reviewer for the positive evaluation of our work and for the very valuable suggestions aiming at its further improvement. When possible, the manuscript has been revised in light of those suggestions and the necessary amendments/corrections have been introduced. In the other cases a detailed answer to the Reviewer’s comment has been provided.
Comment 1: The significance of this work is not clearly mentioned. As authors mentioned, pentafluorophenyl's advantage of the biological applications is already reported. Even though authors want to replace the cisplatin with new compounds but results didn't show a clear advantage of them (no significant improvement of anti-cancer effects).
Answer: Line 371-375: Sentences “The lower cytotoxicity of the new compounds for normal cells in comparison to cancer cells is their undoubted advantage. Even if their cytotoxicity for cancer cells is not radically lower than cisplatin, they can be a reasonable alternative, especially useful in cisplatin resistant therapies. Their good solubility in aqueous solutions is also important, which will allow to "relieve" therapy with toxic solvents.” were added.
Comment 2: Authors mentioned that compounds 2b and 2c have a higher solubility than 2a but it is subjectively mentioned. Please provide the exact number of water solubility of compounds.
Answer: Values of solubility for 2a-c were introduced in Synthesis and Analytical Data.
Comment 3: It is hard to say the tested compounds have lower cytotoxicity on fibroblast cells than cancer cells.
Answer: This part of Results and discussion was rewritten due to Reviewer suggestions.
Line 354-356: Sentences “From the Table 2, we can see that all three complexes show cytotoxicity against fibroblasts cell line (NHDF). Compound 2a and 2b shows higher cytotoxicity to A549 and lower to HeLa and MCF7 cell lines.” were added.
Comment 4: It is interesting that two different compounds showed anti-cancer effects on a specific cell type (2b for A549 and 2c for HeLa). Please provide more explanation of why it happened.
Answer: The mechanism of action is not yet known at this stage of research. However, this is a good starting point for further project. These assumptions can be only elaborated in new, independent, multidisciplinary investigations and presented in subsequent paper due to the length of this manuscript.
Comment 5: As Shen et al. reported (Pharmacol Rev. 2012, 64(3): 706–721), there are several mechanisms on how cells obtained the cisplatin-resistance. If authors explain how new compounds can avoid the cisplatin-resistance mechanism, it would be great to emphasize their contribution not only to pharmaceutical chemistry but also cancer therapy.
Answer: If they could avoid cisplatin resistance, the new compounds would be worth of consideration for cancer treatment, but further studies are necessary and will be considered in the next future. The intimate understanding of the above-mentioned mechanism is complex, but also a valuable object of research and we thank the reviewer for the suggestion.
Following your request, all changes in the manuscript are marked by a yellow background.
Additionally, following the Editor comment, we have partially changed some parts of the manuscript for decrease of a repetition rate (all changes are marked by a blue background).
Round 2
Reviewer 1 Report
All concerns has been clearly addressed by the authors. This reviewer has no additional concern to raise.
Author Response
Cover Letter / Response to Editor
We thank you for your kind consideration of our manuscript.